# Mechanistic studies of a lipase unveil effect of pH on hydrolysis products of small PET modules

Katarzyna Świderek [1] ✉, Susana Velasco-Lozano [2], Miquel À. Galmés[1], Ion Olazabal [3], Haritz Sardon [3], Fernando López-Gallego [2,4] ✉ & Vicent Moliner [1] ✉

Biocatalysis is a key technology enabling plastic recycling. However, despite advances done in the development of plastic-degrading enzymes, the molecular mechanisms that govern their catalytic performance are poorly understood, hampering the engineering of more efficient enzyme-based technologies. In this work, we study the hydrolysis of PET-derived diesters and PET trimers catalyzed by the highly promiscuous lipase B from *Candida antarctica* (CALB) through QM/MM molecular dynamics simulations supported by experimental Michaelis–Menten kinetics. The computational studies reveal the role of the pH on the CALB regioselectivity toward the hydrolysis of bis-(hydroxyethyl) terephthalate (BHET). We exploit this insight to perform a pH-controlled biotransformation that selectively hydrolyzes BHET to either its corresponding diacid or monoesters using both soluble and immobilized CALB. The discoveries presented here can be exploited for the valorization of BHET resulting from the organocatalytic depolymerization of PET.

Plastic recycling is doubtless a realistic solution for the disposal of plastics that contribute to their circularity (reducing the carbon emissions of the plastic industry) and mitigating the plastic landfilling that causes severe pollution. Current technologies to recycle plastics are mainly based on mechanical and chemical methods. Great advances have been done in the chemical recycling of plastics yet the depolymerization processes must be performed using highly pure plastic waste feedstock as otherwise it is difficult to produce highly pure monomers that can be readily transformed into plastics with similar properties to the virgin ones[1,2]. In addition, the selective depolymerization of plastics to polyfunctional molecules opens upcycling pathways for plastic waste. For instance, heterofunctional terephthalic moieties have shown enormous potential as redox-active materials for energy storage devices[3]. In this context, biocatalysis has irrupted to degrade polymers more selectively into monomers that

can be either recycled to the original plastics or upcycled to other building blocks to manufacture products with higher added value[4]. The degradation of poly(ethylene terephthalate) (PET) illustrates how enzyme discovery and engineering have driven the success of biocatalysis on polymer degradation. Since the identification of an efficient PET hydrolase in 2005 from *Thermobifida fusca*[5], many studies have been focused on developing strategies for the enzymatic recycling of PET[6–9]. Remarkably, recent outstanding discoveries have shown how the mesophilic bacteria *Ideonella sakaiensis* 201-F6, can grow in amorphous PET environments, degrading and using it as a carbon and energy source[10,11]. The study of this microorganism revealed a synergic behavior of two enzymes, a PET hydrolase (*Is*PETase) and a mono(2-hydroxyethyl) terephthalate hydrolase (*Is*MHETase). Isolation and characterization of these two biocatalysts confirmed that the former was responsible for the chemical breakage of the polymer into

[1]BioComp Group, Institute of Advanced Materials (INAM), Universitat Jaume I, 12071 Castellón, Spain. [2]Heterogeneous Biocatalysis Laboratory, Center for Cooperative Research in Biomaterials (CIC biomaGUNE), Basque Research and Technology Alliance (BRTA), Paseo de Miramón, 182, 20014 Donostia-San Sebastián, Spain. [3]POLYMAT, Department of Polymer Science and Technology, University of the Basque Country UPV/EHU, Manuel de Lardizabal, 3, 20018 Donostia-San Sebastián, Spain. [4]IKERBASQUE, Basque Foundation for Science, 48013 Bilbao, Spain. ✉e-mail: swiderek@uji.es; flopez@cicbiomagune.es; moliner@uji.es

monomers, principally MHET, while MHETase would catalyze the breaking of this molecule into ethylene glycol and terephthalate (Fig. 1). More recently, Palm et al. have reported kinetic studies for the hydrolysis of BHET and MHET with IsMHETase mutants, showing different activities depending on the specific variants[12]. However, this study did not test the selectivity of the mutants with a BHET and MHET mixture.

We recently studied the reaction mechanism of the PET degradation catalyzed by IsPETase and IsMHETase, as well as by the metagenome-derived leaf-branch compost cutinase (LCC-ICCG variant)[13] by computational methods with predicted rate constants in agreement with the experimental observations[14]. Further analysis of the structural changes of IsPETase induced by PET binding[15], appears to be also in agreement with recent results of Alper and co-workers[16]. The results derived from these computational studies represent the bedrock to understanding the molecular mechanism that rules the performance of PET degrading enzymes. Nevertheless, we poorly understand the mechanism of other well-known ester hydrolases like the highly promiscuous lipase B from Candida antartica B (CALB), which can be a promising protein scaffold for the hydrolysis of polyesters like PET[17].

CALB is a lipase that naturally hydrolyzes triacylglycerides, but it is also capable to catalyze a plethora of chemical processes, including the depolymerization of polyesters such as polylactic acid[18,19], and the hydrolysis of small PET oligomers[20]. Analysis of its active site reveals the presence of a triad formed by Ser105-His224-Asp187 and the presence of an oxyanion hole formed by the residues Thr40 and Gln106 located at the substrate binding pocket. The hydrolytic mechanism of CALB[21] is equivalent to the one proposed for the PET degradation catalyzed by IsPETase, IsMHETase and LCC-ICCG[13]. We know that a suitable protonation state of the CALB active site is fundamental for its

amidase activity[22], yet the effect of pH has never been investigated for PET hydrolysis. Thus, although CALB has been scarcely employed for degrading PET, we envision CALB as a promising chassis to design a highly efficient PETase due to its activity promiscuity to dismantle other polyesters[17], its thermal stability, and our previous computational studies on the catalytic promiscuous activity of CALB[21,23–25]. Furthermore, CALB tolerates organic solvents[26,27] and ionic liquids[28], which may help the downstream and recovery of the PET hydrolytic products. Moreover, Novozyme 435, the commercial immobilized version of CALB is extremely robust and allows its recycling as demonstrated for a wide variety of biotechnological applications[29–31].

In this work we present a computational study supported by experimental data that reveal the unknown role of the pH on the CALB selectivity toward either the full hydrolysis of bis-(hydroxyethyl) terephthalate (BHET) to terephthalic acid (TPA) and two molecules of ethylene glycol (EG) or the partial hydrolysis of the same model substrate to yield mono(2-hydroxyethyl) terephthalate (MHET) and EG (Fig. 1). We exploit the mechanistic insight to perform pH-controlled biotransformation that selectively hydrolyzes BHET to either the diacid or the monoester using both soluble and immobilized CALB. The latter is re-used for up to eight cycles without altering its product profile at two extreme pHs. The potential of this pH-controlled biotransformation is illustrated by the valorization of BHET and small PET oligomers resulting from the organocatalytic depolymerization of PET to yield MHET, an intermediate that is very hard to achieve through chemical methods under mild conditions (i.e., pH 7 and room temperature). Transforming PET waste into monomers different from virgin ones opens a myriad of possibilities to manufacture high-added chemicals and contributes to implementing circular designs in the plastic industry.

## Results
### BHET hydrolysis with selective product formation by pH controlling
First, we studied the effect of pH during the hydrolysis of BHET catalyzed by CALB. To that aim, we performed the time reaction courses for the BHET hydrolysis at different pH conditions and the reaction products were analyzed by UPLC-MS. In all cases, we used high concentrated corresponding buffer to maintain reaction pH along the enzyme hydrolysis. The high ionic strength of the buffer negligibly affects the enzyme activity (Supplementary Fig. 3a). Acidic conditions (pH 5) lead to the hydrolysis of the two ester bonds forming BHET, yielding TPA as the main product upon 8 h (Supplementary Figs. 3 and 4A). In contrast, neutral and alkaline pHs (7 and 9) lead to the hydrolysis of only one ester of BHET, yielding MHET as the only reaction product (Supplementary Figs. 3, 4B and 4C). Surprisingly, the intermediate MHET was a poor substrate for the CALB at pH 9 and 7, but an excellent substrate at pH 5. Blank experiments without the enzyme confirm that the spontaneous hydrolysis of BHET to either MHET or TPA did not occur at pH 5 and 7, while it was significant at pH 9. However, although at pH 9, BHET was spontaneously hydrolyzed to MHET, the presence of CALB accelerates the selective hydrolysis of one of the ester bonds of the substrate. CALB at pH 5 behaves similarly to other esterases capable of hydrolyzing BHET[32], however, this enzyme presents a unique selectivity for the monoesters at alkaline pH. Therefore, these experimental data suggest that the selectivity of CALB toward BHET is significantly affected by the pH of the reaction media. Hence, this pH modulation of CALB hydrolysis of BHET can be exploited for the selective production of either TPA or MHET.

When we determined the Michaelis−Menten kinetic parameters ($K_M$, $k_{cat}$ and catalytic efficiency; $k_{cat}/K_M$) toward BHET and MHET (Table 1, Supplementary Fig. 5), we confirmed the pH dependence of the CALB substrate selectivity. We observed that $k_{cat}$ values are similar for BHET hydrolysis regardless of the pH of the reaction, however, the higher the pH, the lower the $k_{cat}$ for MHET hydrolysis. Regarding

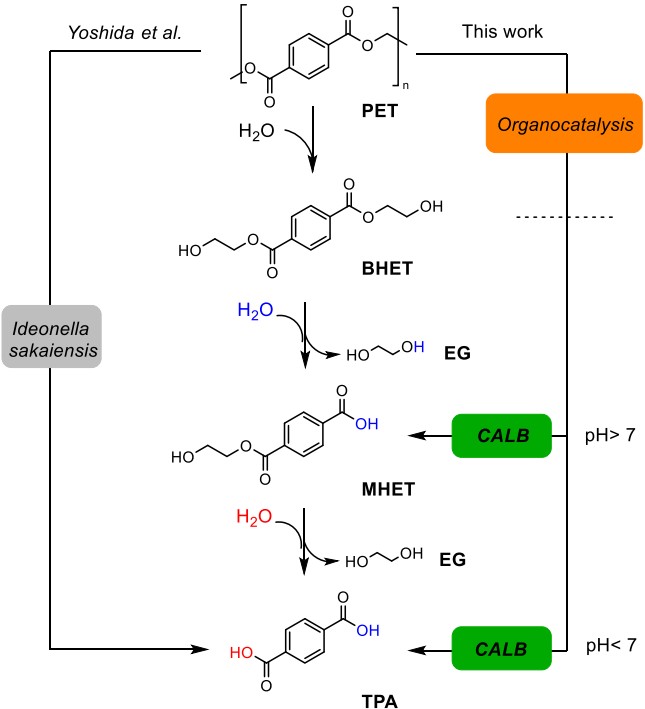

**Fig. 1 | General scheme of the mechanism of biodegradation of PET.** The mechanism of biodegradation of PET to TPA (terephthalic acid) and EG (ethylene glycol) through intermediates BHET (bis-(hydroxyethyl) terephthalate) and MHET (mono(2-hydroxyethyl) terephthalate) catalyzed by PETase and MHETase from *Ideonella sakaiensis* (Yoshida et al.[10], left) and by the synergy of organocatalysts and the lipases from *Candida antartica* B (CALB) (right, this work). The selectivity of the chemo-enzymatic process can be controlled by the reaction pH.

**Table 1 | Michaelis–Menten steady-state parameters of CALB on PET derivatives**

| Substrate | pH | $K_M$ (mM) | $V_{max}$ (U/mg) | $k_{cat}$ (s⁻¹) | $k_{cat}$ / $K_M$ (mM⁻¹ x s⁻¹) | $\triangle G^{act}_{exp}$ |
|---|---|---|---|---|---|---|
| BHET | 5 | 22.5[a] ± 9.6 | 0.94 ± 0.14 | 0.52[a] ± 0.07 | 0.023[b,c] ± 0.010 | 17.8 |
| | 7 | 13.3 ± 4.3 | 1.62 ± 0.27 | 0.89 ± 0.15 | 0.067[d] ± 0.024 | 17.5 |
| | 9 | 61.0[a] ± 25.7 | 1.33 ± 0.45 | 0.73[a] ± 0.25 | 0.012[c,e] ± 0.006 | 17.6 |
| MHET | 5 | 23.8[a] ± 8.8 | 2.27 ± 0.53 | 1.25[a] ± 0.29 | 0.052[b,d] ± 0.023 | 17.3 |
| | 7 | 15.1 ± 3.2 | 0.25 ± 0.02 | 0.14 ± 0.011 | 0.009[e] ± 0.002 | 18.6 |
| | 9 | >100 | n.a. | n.a. | n.a. | na |

Activity assay: Substrate in 10% DMSO and 100 mM buffer at 25 °C. Buffer for pH 5 was sodium acetate, for pH 7 was sodium phosphate and for pH 9 was sodium bicarbonate. n.a. not assessed because it did not reach substrate saturation. [a]Values were extrapolated out of the upper limit of 20 and 50 mM of BHET and MHET as higher concentrations cannot be achieved under the activity assay conditions due to the solubility limit of both substrates.[b,c,d,e] Means with a different letter are significantly different ($p < 0.04124$). Data were determined from three independent replicates and analyzed by ordinary one-way ANOVA with Tukey's multiple comparison test.

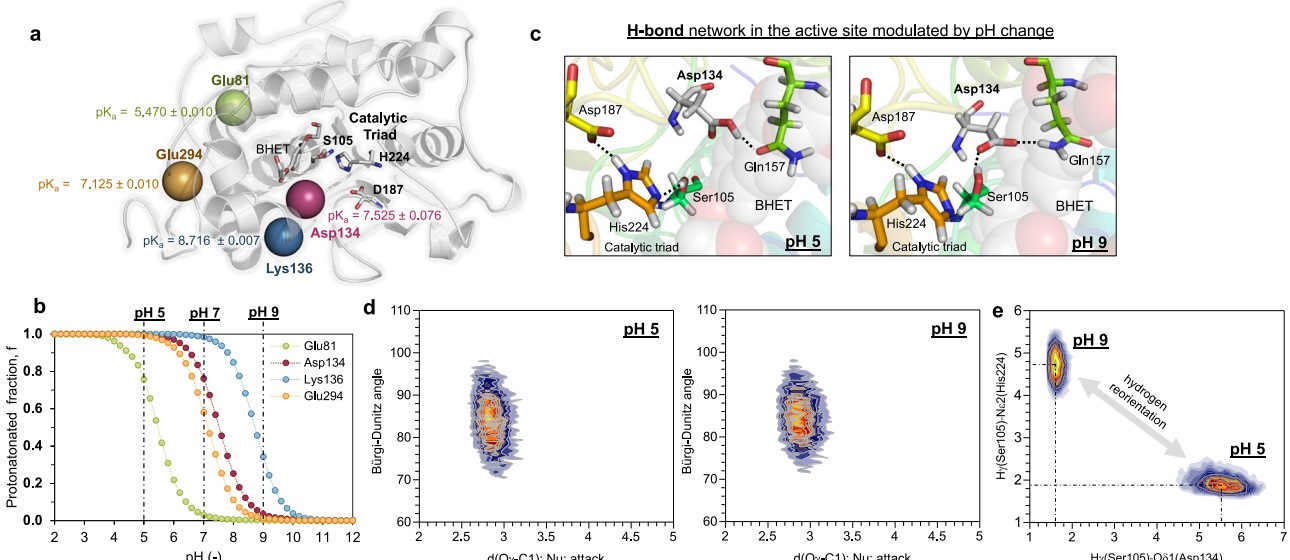

**Fig. 2 | pH effect on the CALB:BHET reactant complex conformations. a** Detail of the structure of CALB with the computed pKa values for uncertain titratable residues. Positions Cα of key residues in the protein are shown as spheres. **b** Computed titration curves for key amino acid residues of CALB generated by constant-pH neMD/MC simulations. **c** Detail of the interactions between Asp134 and Gln157 and Ser105 created in the active site in the CALB: BHET reactant complex at pH 5 and 9. **d** Population analysis of geometrical parameters (Nu⋯Csp²=O Bürgi-Dunitz angle in degrees, and Nu⋯Csp² distance in Å) defining the nucleophilic attack of Ser105 to C1 atom of BHET. **e** Population analysis defining the relative position of Hγ of Ser105 and its activator His224, and of Hγ of Ser105 and Asp134, at pH 5 and 9.

substrate affinity, the $K_M$ of CALB toward both substrates, BHET and MHET, follows the same trend; $K_M$ (pH 9) > (pH 5) > (pH 7). Therefore, CALB hydrolyzes BHET more efficiently at pH 7 due to the combination of the highest $k_{cat}$ and the lowest $K_M$, unlike the hydrolysis of MHET which is extremely slow ($k_{cat}$ of unless 1 order of magnitude lower) at pH higher than 5, which precludes the formation of TPA at the neutral and alkaline pH herein tested. Remarkably, CALB hydrolyzes BHET at pH 7 as efficiently as MHET at pH 5, however, at pH 5 the rate-limiting step was the hydrolysis of the diester. Hence, CALB prefers the diester substrate at neutral pH while it does the monoester at an acid pH, supporting the fact that TPA resulting from the two-consecutive hydrolysis is accumulated only when the reaction pH is acidic.

**MD simulations to estimate the protonation states of CALB:-substrate complexes at different pH**

According to the experimental data shown in Table 1 and Supplementary Fig. 3, CALB hydrolyzes the two esters bonds of BHET to yield TPA and 2 molecules of EG under acidic conditions (pH 5), however, this enzyme only hydrolyzes one of the esters of BHET to yield the monoester MHET under alkaline conditions (pH 9). To find out the origin of this phenomenon, and keeping in mind the relevance of

the protonation state of the titratable residues in the intra-molecular interactions and conformations of proteins, we first performed an analysis of the pKa values of selected residues by using the empirical program PropKa ver. 3.0 3[33] and by getting the full titratable curves based on constant-pH non-equilibrium molecular dynamic and Monte Carlo (neMD/MC) simulations[34,35]. According to the results derived from both methods, residues Glu81, Asp134, Lys136 and Glu294 must be protonated at pH 5, whereas they are all mostly deprotonated at pH 9 (Fig. 2a, b, and Supplementary Figs. 11 and 12). This must be obviously related to local conformation changes of the protein (Fig. 2c, d, and Supplementary Figs. 13–16), but apparently without significant effects on the efficiency of CALB to hydrolyze BHET, according to our experimental kinetics (see Table 1). Interestingly, when making the comparison between pH 5 and 7, the main difference appears in the Glu81, which is protonated at pH 5 but deprotonated at pH 7. Lys136, Glu294 and Asp134 do not change their protonation state from pH 5 to 7. Experimental data thus suggest that the protonation state of Glu81, which is located more than 20 Å from the substrate, negligibly affects the performance of CALB toward the first hydrolysis of BHET.

The detailed analysis of geometries of the CALB:BHET at pH 5 and 9 after equilibration of the systems by MD simulations with the proper

protonation states of the titratable residues showed significant differences. First, it is confirmed that the different protonation state of Glu81 makes some changes in the interaction network with the neighbor residues by changing the relative orientation of their side chains (Supplementary Fig. 15), an effect that is also observed in the case of Glu294 and Lys136 (Supplementary Figs. 13 and 14). More importantly, it appears that the protonation state of Asp134 triggers the rearrangement of the H-bond interaction network between BHET and the CALB active site (Figs. 2c, d and Supplementary Fig. 16). Thus, the carboxylic group of Asp134 at pH 5 forms a hydrogen bond with the carbonyl group of Gln157 that establishes a hydrogen bond network with Thr138 through the -NH groups of the backbone, and with Thr40. These interactions contribute to the formation of the oxyanion hole that anchors the substrate to the active site. In contrast, the ionic form of Asp134 at pH 9 creates a completely different hydrogen bond network, now established with the -OH group of Ser201, Thr138 and Gln157 through a water molecule (Fig. 2c). These conformational changes do not significantly affect the distance and angle of the nucleophilic attack between the reactive Ser105 and the substrate (Fig. 2d). Nevertheless, there is an effect on the relative position between Ser105 and His224, that can compromise the proton transfer required to activate the catalytic Ser105 (Fig. 2e). Thus, while the average distance between the proton of Ser105 and the Nε2 atom of His224 is ca. 2.0 Å at pH 5, this value increases to ca. 4.5 Å at pH 9. In contrast, the distance between the deprotonated Asp134 and Ser105 is reduced from 5.5 to 1.9 Å. These changes in the conformation of the active site do not provoke a diminution of the catalytic activity of CALB in degrading BHET, as experimentally demonstrated, thus suggesting a plasticity of the enzyme that allows "conformational" promiscuity.

Regarding the substrate, it is expected that the protonation state of hydroxyl groups of BHET did not change in the full range of different working pHs, and the same could be applied to the MHET since the $pK_a$ of the acidic groups of MHET ranges from 3.5 to 4.8 depending on the substrate configuration within the active site and its local environment[36]. However, keeping in mind that this range is close to pH 5, a non-negligible population of the available MHET substrate could be protonated under acidic conditions. Hence, we assumed the carboxylic group of MHET is negatively charged at pH 9 (MHET$^{(-)}$), but we tested a protonated and negatively charged state at pH 5 (MHET and MHET$^{(-)}$, respectively). Results of MD simulations performed at the two limit pHs with the different protonation states of the previously commented residues and MHET are shown in Fig. 3 and Supplementary Movies 1 and 2, where the time evolution of the distance between the center of mass of MHET and the center of mass of the oxyanion hole of the active site at different pHs is shown. The results have shown the diffusion of MHET to the bulk at pH 9, but not at pH 5, independent of whether the substrate is considered neutral or negatively charged.

Then, we monitored the interactions between the substrate (BHET or the negatively charged MHET) and the active site residues at both pHs (Supplementary Figs. 17 and 18) to carry out a deeper analysis. The interaction energies between MHET and CALB decomposed by residues show that the deprotonated Asp134 at pH 9 promotes the repulsion of the substrate in the Michaelis complex, which is not observed in the CALB:BHET complex. The interaction energy between Asp134 and the substrate is much more negative (more stabilizing) at pH 5 than at pH 9. The negative charges located on Asp134 and MHET promote the repulsion of the substrate from the active site at pH 9. In contrast, acidic conditions (pH 5) let the substrate form part of the interaction network that sustains a productive conformation of the active site. The differences in the interactions of the rest of the residues at the two tested pHs are negligible. This analysis of the interaction energies calculated at the two different pH values agrees with the residence time of the substrate in the active site during unconstraint MD simulations, as discussed above. Thus, MD analysis suggests a relevant role of the Asp134 ultimately to bind MHET. In summary, a negatively charged Asp134 prevents the formation of the Michaelis complex for the hydrolysis reaction of MHET to TPA and EG, explaining why the hydrolysis of the intermediate MHET is precluded at pH 9, which halts the enzymatic process in the first ester hydrolysis of BHET accumulating MHET. In order to provide additional support to this hypothesis, we have carried out MD simulations of the MHET in the active site of an Asp134Ala variant. The results show how the repulsion established between the negatively charged Asp134 and deprotonated MHET$^{(-)}$ at pH 9 in the wild-type enzyme vanishes in the new variant. Mutation of Asp134 to Ala restores the favorable interactions between MHET$^{(-)}$ and the active site, and therefore the substrate remains in the pocket during the unconstrained MD simulations, thus confirming the role of the negatively charged Asp134 in preventing the binding of MHET at pH 9 (Supplementary Fig. 19). Remarkably, our computational results align with the experimental Michaelis–Menten parameters that show how the $K_M$ toward MHET at pH 5 was significantly lower (23.8 ± 8.8 mM) than at pH 9 (>100 mM) (Table 1). In fact, CALB did not reach the substrate saturation plateau when MHET was used as substrate and the reaction was performed at pH 9.

## QM/MM study of the hydrolysis of BHET catalyzed by CALB

The observed dramatic unfavorable effect on the binding of MHET at pH 9 indicates that the substrate will hardly bind the active site at this pH, and the virtually equivalent activation free energies derived from the experimentally determined rate constants of the hydrolysis of BHET and MHET at pH 5, as listed in Table 1 (17.8 vs 17.3 kcal·mol$^{-1}$), suggests we focus on exploring the free energy landscape corresponding to the hydrolysis of BHET catalyzed by CALB at pH 5 and pH 9. However, in order to confirm the hypothesis of equivalent reaction

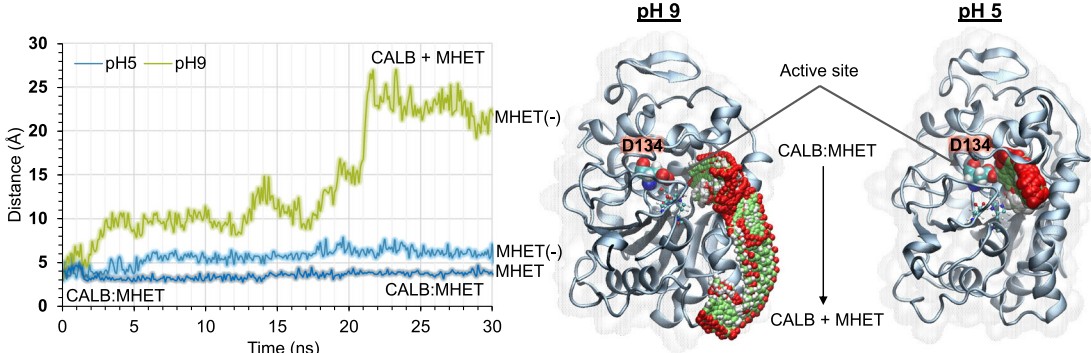

**Fig. 3 | Stability of CALB:MHET complex systems.** Left panel: time dependence evolution of the distance between the center of mass of the substrate and the oxyanion hole along the unconstraint MD simulation in the CALB:MHET reactants complex at pH 5 (light blue and dark blue lines for negatively charged MHET(−) and neutral MHET, respectively) and pH 9 (green line). Right panel: superposition of structures generated along the MD simulations at pH 5 and pH 9.

mechanisms for the hydrolysis of BHET and MHET, free energy surfaces (FESs) were explored not only with BHET at the two pH values, but with MHET at pH 5, conditions under which the reaction is feasible as suggested by the experimental data (Supplementary Figs. 20 and 22). First of all, the results confirm that the hydrolysis, independently of whether the substrate is BHET or MHET, takes place in four steps, as originally predicted based on the previous studies of the hydrolase and amidase activity of CALB performed in our laboratory[22,23,25]. The FESs of MHET hydrolysis (Supplementary Fig. 22) are virtually the same as those for the hydrolysis of BHET (Supplementary Figs. 20 and 21). Focusing on the hydrolysis of BHET, the first two steps involve the formation of the acyl-enzyme complex and the formation of an EG molecule, where the deprotonated hydroxyl group of the Ser105 activated by His224 attacks the carbonyl of BHET. At this point, EG is formed as leaving group and the acyl-enzyme complex is ready for the hydrolysis step. Next, a water molecule activated by the catalytic His224 attacks the carbonyl carbon forming a tetrahedral intermediate. Finally, the His224-assisted hydrogen reshuffling triggers the resolution of the acyl-enzyme complex that is followed by the formation of MHET as a final product to regenerate the CALB active site ready to start a new catalytic cycle. A list of coordinates for the structures of all states appearing along the reaction at the two pH values optimized at QM/MM level, with the QM atoms treated at density functional theory (DFT) level, is provided in Supplementary Tables 5 and 6.

Figure 4 shows the free energy profiles for the hydrolysis of BHET at pH 5 and 9 (see Supplementary Table 4 for a list of relative energies). Here, we can observe that the kinetics of the first ester hydrolysis of BHET is limited in both cases by the hydrolysis step. In particular, the rate-limiting step is determined by TS4, giving total activation free energy values of 18.0 and 17.1 kcal·mol⁻¹ for the hydrolysis at pH 5 and 9, respectively. These predicted activation Gibbs free energies, which can be considered indistinguishable considering the uncertainty associated with the computational method, agree with the experimental values, 17.8 and 17.6 kcal·mol⁻¹, that can be derived by applying the Transition State Theory[37] to the previously measured rate constants (see Table 1). Interestingly, the energetic data derived from the FESs corresponding to the hydrolysis of MHET render a value of the free energy of activation of the rate-limiting step of 18.3 kcal·mol⁻¹, very close to those for the hydrolysis of BHET and in agreement with the indistinguishable kinetic data. The mechanistic and kinetic description of the process agrees with previous studies on the reactivity of CALB[23,38] and *Bacillus subtilis* Bs2[22,25] where we noticed that the rate-determining step of the overall reaction is controlled by the hydrolysis step. It is interesting to point out that, in agreement with the geometrical analysis of the Michaelis complexes discussed above,

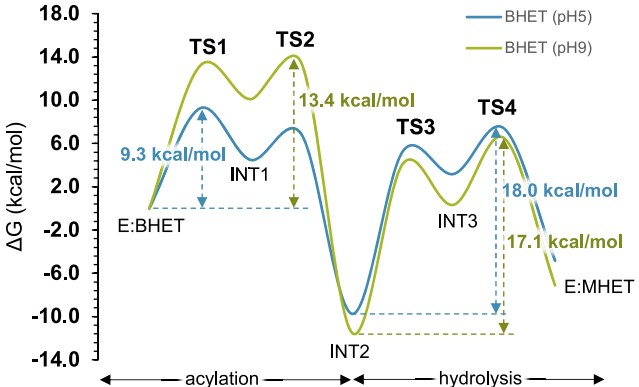

**Fig. 4 | Free energy profiles of the hydrolysis of BHET.** Free energy profile of the hydrolysis of BHET at pH 5 (blue line) and pH 9 (green line), catalyzed by CALB and computed at M06-2X/OPLS-AA level. All profiles include ZPE corrections.

the free energy barriers for the acylation step under alkaline conditions are clearly higher than those under acidic conditions (13.4 vs 9.3 kcal·mol⁻¹). This means that, despite the conformational promiscuity of the active site, the relative orientation of the substrate and the active site at pH 5 is more reactive than the one at pH 9. Obviously, because the acylation step is not determining the kinetics of the overall process, this difference in the energy barriers is not reflected in different $k_{cat}$ values ($0.52 \pm 0.07$ vs $0.73 \pm 0.25\ s^{-1}$ at pH 5 and 9, respectively, as shown in Table 1). However, this computational prediction supports the experimentally determined lower $K_M$ at pH 5 than a pH 9 ($22.5 \pm 9.6$ vs $61.0 \pm 25.7$ mM, respectively, as shown in Table 1).

As observed in our previous studies on reactions catalyzed by CALB[23,25], the product of the acylation step in all the two reactions (INT2 in Fig. 4) is stabilized by comparison with the reactant complex (−9.7 and −11.5 kcal·mol⁻¹). This effect may be partially due to the electrostatic stabilization of the acyl-enzyme complex. The development of higher densities of charges in key atoms of the substrate in intermediate 2, by comparison with the reactant complex, can explain a higher stabilization of this state, at both pHs (Supplementary Table 7).

## pH-controlled CALB hydrolysis of BHET and PET oligomers resulting from the chemical depolymerization of PET

PET polymer was depolymerized following the method previously described by Jehano et al. [39] NMR spectrum revealed that the major depolymerization product was BHET (hitherto named crude BHET). Encouraged by the performance of CALB to selectively hydrolyze only one or the two ester bonds of BHET depending on the reaction pH, we hydrolyzed the crude BHET at pH 5 and 9 to selectively yield TPA and MHET, respectively. The resulting products were analyzed by UPLC-MS and ¹H-NMR. Figure 5a, b shows the reaction time course of crude BHET at pH 5 and 9. As expected, CALB transformed the BHET crude into TPA in just 8 h negligibly accumulating the intermediate MHET. In contrast, at pH 9, we observed that the only product formed was MHET as its enzymatic hydrolysis was precluded under these conditions. NMR studies confirmed that at pH 5 the only hydrolysis product was TPA, while MHET appears when the reaction was performed at 9 with traces of TPA though (Supplementary Figs. 6 and 8). We found the same selectivity switch when a PET trimer coming from the depolymerization process was submitted to CALB hydrolysis at pH 5 and pH 9. Under acidic conditions, TPA was majorly detected as a monomer unlike alkaline conditions where the major product was MHET (Supplementary Fig. 9). The titers of MHET and TPA (1–4 mM) using the PET trimer as starting material were lower than using BHET due to the poor solubility of the PET trimer in the reaction media (Supplementary Fig. 9A–D). Hence, the accessibility of CALB to the unsoluble PET trimer is limited, which explains the lower hydrolysis yield. Upon 24 h of reaction, ¹H-NMR analysis reveals that the trimer is majorly detected in the unsoluble fraction regardless of the reaction pH, unlike the TPA (at pH 5) and MHET (at pH 9) that are selectively accumulated in the soluble fractions (Supplementary Fig S9E–H). The presence of ethylene glycol in the NMR also supports the hydrolytic activity of CALB over the PET oligomer. Hence, the selective production of either TPA or MHET from a depolymerization crude opens an avenue to revalorize plastic wastes through chemo-enzymatic methods where chemical catalysts depolymerize plastic polymer into small oligomers that are valorized by the action of the enzyme into monomers that can be upcycled into other products. For example, TPA can be used as a carbon source for the microbial fermentation of polyhydroxyalkanoates (PHAs) and polycathecols[32,40–42] and the chemical production of virgin-like PET. Alternatively, MHET can be further upcycled into new materials taking advantage of the orthogonal functionality of the hydroxy and carboxylic groups. In fact, the complementary functionality of these monomers eases their further chemical transformations, increasing their purity and reducing the

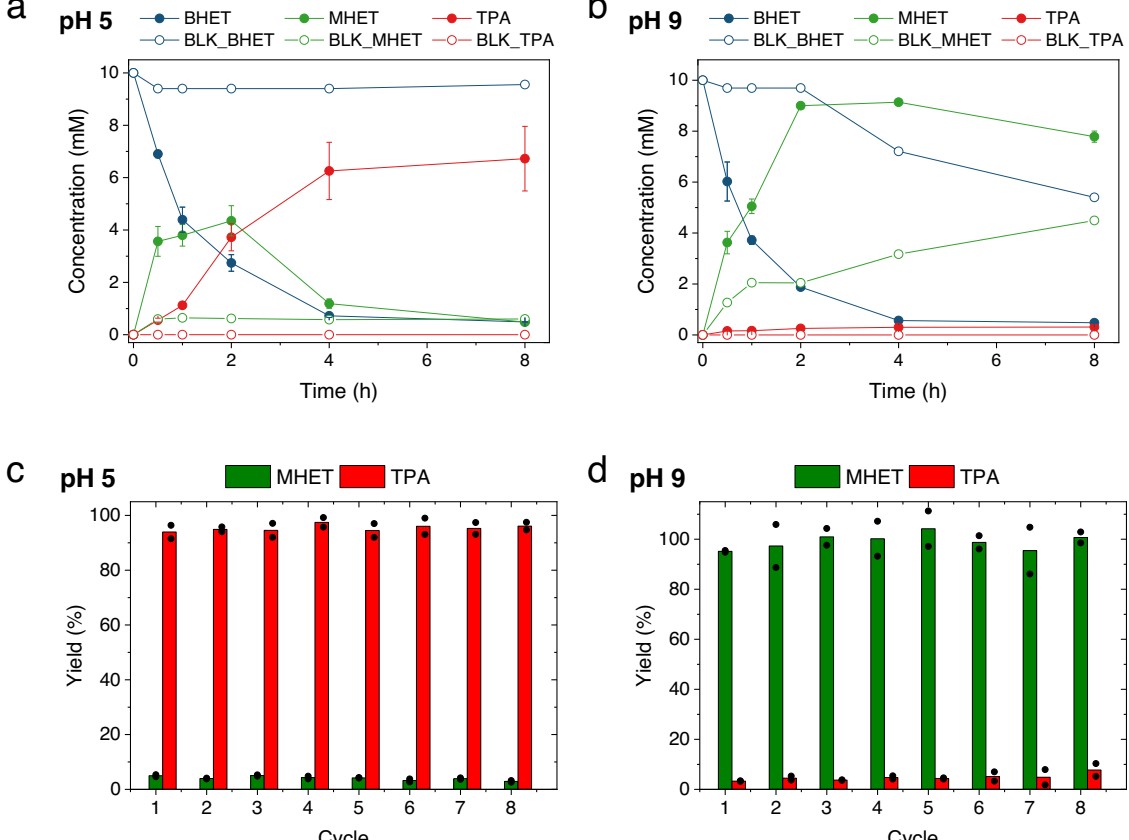

**Fig. 5 | pH dependence of CALB hydrolysis of crude BHET.** Time courses of the pH-controlled hydrolysis of BHET crude from chemical PET depolymerization at **a** pH 5 and **b** pH 9. Filled circles represent enzymatic reaction products, while empty circles represent reaction products in the absence of the enzyme. In all cases, 36 μL CALB (5.12 mg/mL) was mixed with 1800 μL of 10 mM BHET in 100 mM buffer containing 10% DMSO and incubated at 25 °C. Sodium acetate and sodium bicarbonate were used as a buffer for the reaction at pH 5 and 9, respectively.

Recycling of immobilized CALB during BHET hydrolysis at **c** pH 5 (one cycle corresponds to 24 h), and **d** pH 9 (one cycle corresponds to 4 h). In all cases, 9 mg Novo435® was mixed with 500 μL of 10 mM BHET in 100 mM sodium bicarbonate containing 10% DMSO and incubated at 25 °C. Data are the mean value of $n = 2$ independent measurements and error bars mean the standard deviation of the independent duplicates.

number of potential side reactions without generating a significant amount of waste.

### pH-controlled hydrolysis of BHET catalyzed by immobilized CALB

Motivated by these excellent results obtained with soluble CALB, we investigated if the pH control of the hydrolysis was maintained when using an immobilized formulation of CALB. To that aim, we used the commercial preparation of CALB immobilized on polymethacrylate beads (Novo435®). As expected, MHET could only be obtained when the reaction was performed at pH 9, while TPA was achieved operating the reaction at pH 5 (Fig. 5c, d). Therefore, the immobilization of CALB does not affect its reaction selectivity under different pH conditions. Moreover, the immobilized preparation maintains both, its original activity and reaction product profile during 8 reaction cycles. These results point out the suitability of CALB as an industrial biocatalyst for the degradation of PET oligomers with unprecedented substrate selectivity by simple pH control.

### Discussion

In summary, we propose a selectivity switch for the formation of either MHET or TPA from BHET by CALB by modulating the pH conditions, through classical and QM/MM MD simulations combined with experimental Michaelis−Menten kinetics. Our results show how the ionization state of CALB under acidic conditions forms a neutral

hydrogen bond network that enables the binding of both the uncharged initial substrate BHET and the negatively charged intermediate MHET, explaining its capacity to perform the double hydrolysis of the former under acidic conditions. In contrast, the MHET anion together with the ionization state of several residues of the protein, including Asp134 at pH >7, destabilizes the CALB:MHET complex avoiding its hydrolysis and explaining the selective hydrolysis of one out of the two ester bonds of BHET. The exploration of the QM/MM free energy landscape of the hydrolysis of BHET, at pH 9 and pH 5, allows for obtaining a detailed description of this enzyme-catalyzed multistep chemical reaction. Conformational changes taking place in the active site as the result of the different protonation states of titratable residues influence the barrier of the acylation step, but not the rate-limiting hydrolysis step. Computational results were supported by experimental Michaelis−Menten kinetics of CALB at different pH conditions, with activation free energies of the rate-limiting step following the same trend as the experimentally determined rate constants. Finally, we exploited the pH-controlled hydrolysis to transform BHET and PET trimers resulting from the chemical depolymerization of PET into individual monomers such as either TPA or MHET depending on the reaction pH. We confirmed that the product profile at the two different pHs was maintained even using an immobilized preparation of CALB. The results we present here open avenues for the exploitation of CALB in polymer recycling and upcycling. The herein-discovered selectivity of CALB allows us to use, but also re-use,

the same biocatalysts to prepare either diacids or monoacids from diesters and PET trimers, and eventually from polyesters. Biocatalytic tools, such as the one presented in this work, forecast pathways to recycle, but also upcycle, plastic wastes upon the chemical depolymerization process. Furthermore, we forecast that more chemo/enzymatic approaches like the one herein presented will be deployed in the short term to make plastic recycling industrially appealing.

## Methods

### Materials

Soluble lipase B from *Candida antarctica* Lipozyme® and its immobilized preparation Novozym435® were acquired from Novozymes (Denmark). Substrates and reaction products, bis(2-hydroxyethyl) terephthalate (BHET), terephthalic acid (TPA); chemicals and solvents, 2-4-morpholinepropanesulfonic acid (MOPS), sodium acetate anhydrous, sodium phosphate, sodium bicarbonate, dimethyl sulfoxide, acetonitrile were acquired from Sigma-Aldrich (St. Louis, IL). Precision plus proteinTM standards and Bradford reagent were acquired from BIORAD. All other reagents and solvents were analytical grade or superior.

### Protein quantitation

The protein content of soluble *Candida antarctica* Lipozyme® was determined by SDS-PAGE gel electrophoresis by employing a broad range of protein molecular weight markers (Promega) (Supplementary Fig. 1).

### Enzyme activity measurements

Enzyme activities were spectrophotometrically measured in transparent 96-well microplates, employing a Microplate Reader Epoch 2, BioTek® with the software Gen5.

Lipase activity was indirectly monitored by the decrease in the pH triggered by the TPA formation from total BHET hydrolysis. Briefly, 200 µL of a reaction mixture containing BHET (at the required concentration), 10% acetonitrile, and 0.25 mM *p*-nitrophenol in 25 mM MOPS buffer at pH 7.2 were incubated with 5 µL of enzymatic solution or suspension (properly diluted) at 30 °C. The decrease in the absorbance of *p*-nitrophenol (pH indicator) at 410 nm was recorded. One unit of activity was defined as the amount of enzyme that was required to produce 1 µmol TPA per minute at the assayed conditions.

### Depolymerization of PET

For this, 0.5 g of PET flakes were depolymerized by 20 eq. ethylene glycol employing 0.5 eq of TBD:MSA as the catalyst. A 26 mL vial equipped with a magnetic stirrer was used for all the reactions. The depolymerizations were conducted under atmospheric pressure at 180 °C for 2 h until the complete disappearance of any residual PET. Reagents and catalysts were loaded in the glovebox, under a nitrogen atmosphere, before sealing the flask and immersion in a pre-heated oil bath. After reaction completion, the crude was cooled down to room temperature, and a large excess of distilled water was added. The resulting mixture was vigorously stirred before being filtered to separate ethylene glycol, catalyst and main product from oligomers which are insoluble in water. The aqueous transparent filtrate was stored in a refrigerator at 4 °C overnight. After this, white needle-like crystals were formed in the solution, which were filtrated and dried. 1H NMR characterization revealed that the collected crystals were highly pure bis(hydroxyethyl)terephthalate (BHET) monomer with characterizing data in accordance with commercially-supplied BHET (Supplementary Fig. 2).

### BHET and PET trimer hydrolysis

Either soluble or immobilized enzymes were placed inside a 2 mL Eppendorf tube containing a reaction mixture (0.5 or 1.8 mL, as

indicated) consisting of 10 mM BHET or PET trimer, 10% DMSO in 100 mM buffered aqueous solution assuring the pH maintenance during the whole hydrolysis reaction without detrimental effects on CALB activity (sodium acetate for pH 5, sodium phosphate pH 7, sodium bicarbonate for pH 9). Reactions were incubated at 25 °C at 250 rpm. The reaction course was monitored by withdrawing samples at periodic intervals using cellulose membrane tangential units (10kDa MWCO) to remove the enzyme. Then, the samples were analyzed by chromatographic methods and $^1$H-NMR. In the case of the PET trimer, due to its solubility issues, the reaction crude was centrifuged and the solid (precipitate) and the aqueous (supernatant) fractions were separated. The solid fraction was solubilized in pure DMSO and the soluble fraction was passed through the cellulose membrane tangential units. The concentrations of substrate (BHET), intermediate (MHET) and product (TPA) were determined by UPLC-MS analysis at different time points. Prior UPLC-MS analysis samples were diluted 5 times in 100 mM acetate buffer to avoid BHET autohydrolysis at alkaline pHs. MHET and TPA yields (%) were calculated MHET or TPA concentration at each reaction time divided by the initial concentration of BHET and multiplied by 100.

### $^1$H-NMR analysis

Prior NMR analysis reaction samples were filtered through cellulose membrane tangential filtration units (10 kDa MWCO) and diluted 1.1 times with deuterated water. $^1$H Nuclear Magnetic Resonance (NMR). 1H-NMR spectroscopic measurements were carried out on a Bruker Advance 400 (400 MHz) spectrometer. If needed, to improve spectrum quality water signal was suppressed by means of presaturation experiments.

### UPLC-MS analysis

Prior UPLC-MS analysis of reaction samples were properly diluted (typically 10–20 times in acetonitrile/water mixture 30:70). The analysis was performed in an Acquity UPLC system using an Acquity BEH C18 column (100 × 2.1 mm, 1.7 µm) from Waters (Milford, MA, USA) and equipped with a photodiode array detector (PDA). The gradient elution mobile phases were A: 0.1% Formic acid in water, and B: acetonitrile. The gradient method was: 0–1 min, isocratic at 90% A; 1–7 min, gradient to 50% A; 7–8.3 min, isocratic at 50% A; 8.3–8.5 min, gradient to 90% A, 8.5–10 min isocratic for stabilization at 90% A. The UV detector wavelength was set at 245 nm and the injection volume was 2 µL. The total run time was 10 min with a flow rate of 300 µL·min$^{-1}$. Retention times were: TPA 1.97 min, MHET 2.92 min and BHET 3.41 min.

The mass spectrometry detection was carried out using a time-of-flight mass spectrometer (ESI-TOF) LCT Premier XE from Waters (Milford, MA, USA) with an electrospray ionization source, working in positive/V mode. The MS range acquired was between m/z 50–1.000. The capillary and cone voltages were set at 3.000 and 50 V, respectively. The desolvation gas temperature was 300 °C and the source temperature was 120 °C. The desolvation gas flow was set at 600 L·h$^{-1}$ and cone gas flow was set at 50 L·h$^{-1}$. For quantification and data analysis, Masslynx v4.1 software was used (Waters, Milford, MA, USA). The detection was carried out in positive and negative ions modes, monitoring the most abundant isotope peaks from the mass spectra. BHET (Mw: 254.24) and MHET (Mw: 210.18) were detected as protonated molecules with m/z 255 and 211 respectively (positive ion mode), whereas TPA (Mw: 166.13) was detected as a deprotonated molecule with m/z 165 (negative ion mode).

### Computational model setup

Wild-type *Candida antarctica* Lipase B (CALB) initial geometry was taken from PDB structure 1TCA[43]. Three systems were prepared by assigning the protonation state of titratable residues at pH equivalent to 5 and 9 in agreement with pKa values determined by getting the full

titratable curves based on constant-pH hybrid neMD/MC simulations, as implemented as a Tcl plugin, *namdcph*, for use in conjunction with NAMD ver. 2.12[44]. The substrates BHET and MHET, in their neutral and negatively charged state (MHET and MHET$^{(-)}$, respectively), were manually built in the active site, with the oxygen of the carbonyl properly placed in the oxyanion hole formed by Thr40 and Gln106. Afterward, all missing hydrogen atoms were added to the structure, and the system was solvated into a $100 \times 80 \times 80$ Å$^3$ pre-equilibrated box of TIP3P[45] water molecules, and counterions were added to neutralize the systems.

After initial energy minimizations, the systems were heated to 303 K with a 0.1 K temperature increment and equilibrated during short (100 ps) NPT MD simulations, followed by non-accelerated classical 100 ns NVT MD simulations with AMBER force field[46], as implemented in NAMD software. During the 100 ns of NVT MD simulations, all atoms were free to move within periodic boundary conditions (PBC) and cut-offs for nonbonding interactions with an internal cut-off of 14.5 Å and an external of 16 Å. In order to avoid diffusion of the substrate, a restraint in the position between the oxygen of the carbonyl and the oxyanion hole was applied. The constant temperature was maintained using a Langevin thermostat[47]. See Supplementary Methods for more details.

### Free energy surfaces (FESs)
FESs were obtained, in terms of two-dimensional potential mean force (2D-PMF)[48], for every step of the reaction using the Umbrella Sampling (US) approach[48,49] combined with the Weighted Histogram Analysis Method (WHAM)[50], at the two selected pH. Structures obtained in previously computed PESs were used as starting points for the MD simulations in every window. Due to the cost of the sampling of the system at every window, the PMFs were initially generated with the QM region treated at the AM1[51] level. The OPLS-AA[52] and TIP3P classical force fields were used to treat the protein and the solvent water molecules, respectively, as implemented in the fDynamo library[53]. Then, in order to improve lower quality results associated with the low-level semiempirical calculations, high-level two-dimensional spline function corrections were applied using the M06-2X density functional[54], with the standard 6-31+G(d,p) basis set, as implemented in Gaussian 09 program[55]. Structures of key states involved in the reactions were optimized at M06-2X/MM level. See Supplementary Methods for details.

### Reporting summary
Further information on research design is available in the Nature Portfolio Reporting Summary linked to this article.

## Data availability
The data that support the findings of this study are available within the main text and Supplementary Information. The protein structures used in this work can be found in the Protein Data Bank with the identifier "1TCA". Source data are provided with this paper and available through Zenodo [https://doi.org/10.5281/zenodo.7919733]. Source data are provided with this paper.

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

## Acknowledgements

This work was supported by the Spanish Agency of Research (AEI) (ref. PID2021–123332OB-C21, PID2021–124811OB-I00 and PID2019–107098RJ-I00), the Generalitat Valenciana (PROMETEO, with ref. CIPROM/2021/079, and SEJI/2020/007), Universitat Jaume I (UJI-A2019-04 and UJI-B2020-03). K.Ś. thanks Ministerio de Ciencia e Innovación and Fondo Social Europeo for a *Ramon y Cajal* contract (Ref. RYC2020-030596-I) and a European Cooperation in Science & Technology COST Action (ref. CA21101). This work was partially performed under the Maria de Maeztu Units of Excellence Program from the Spanish State Research Agency Grant MDM-2017-0720. The authors acknowledge the computational resources founded by the Spanish Ministry of Science–European Regional Development Fund (REF: EQC2019-006018-P) installed at Universitat Jaume I. We thank Dr. Grajales for his assistance in the analysis of UPLC-MS samples.

## Author contributions

K.Ś., F.L.G. and V.M. designed the study. K.Ś. and M.A.G. performed all the computer simulations. S.V.L. did most of the experimental work. I.O. and H.S. produced the BHET and PET trimers from organocatalytic depolarization of PET and interpreted the NMR data. K.Ś., S.V.L., M.A.G., F.L.G. and V.M. analyzed the results and contributed to writing the manuscript.

## Competing interests

The authors declare no competing interests.
