## [Peer Review File · Nature Communications]

Mechanistic studies of a lipase unveil effect of pH on hydrolysis products of small PET modulesReviewers' comments:

Reviewer #1 (Remarks to the Author):

The manuscript by Świderek et al. is centered on the coherent topic of biotechnological-assisted plastic recycling. The study is focused on *Candida antarctica* lipase B (CalB) and describes its ability to degrade BHET (obtained from chemical degradation of PET) into either MHET (monoacid) or TPA (diacid). The authors describe a selectivity switch with respect to formation of either MHET or TPA from BHET by CalB at acidic or basic pH, and use computational tools (MD and hybrid QM/MM techniques) to shed mechanistic light on the observed switch. I find the topic of the manuscript interesting and the computational work seems well performed. Still, I have concerns:

Major:

*Novelty: The authors make arguments that their study would enable isolation of either TPA or MHET, with MHET being a difficult to access intermediate by chemistry. In fact, a report by Eugenio et al. have already described the use of hydrolases for efficient biosynthesis of MHET from BHET (10.1016/j.jbiotec.2022.08.019). Likewise, specificity switch for either MHET or BHET as substrate is known (<https://doi.org/10.1038/s41467-019-09326-3>). In the light of these previous report, my opinion is that the present contribution would be more suitable for a specialized journal.

*Experiments and kinetics. What is the solubility of MHET, BHET and TPA under the experimental conditions, and how is the solubility of the compounds affected by the different pH-values used? A hint to the fact that solubility might be an issue is that NMR shows formation of TPA, whereas this was not observed by UPLC-MS, and I assume the workup was different for these methods? (did not find enough details in the experimental section to fully evaluate/understand how the experiments were performed)

*Figure 5c: how was the "yield" quantified?

*What is the impact of ionic strength on the observed results?

*Protonation states: The authors state that MHET is negatively charged at pH 5 and that its pKa resides between 3.5-4.8. If the pKa is close to 5, it means that half of the available substrate would be protonated under the acidic conditions. How is this accounted for in the study? Also, how did the authors take into account the fact that the catalytic histidine will change its protonation state in the pH range studied, and how would that impact the selectivity?

*Fig S6, crude NMR. What are the unassigned peaks?

Minor and editorial comments:

*please use italics for kinetic constants ("*k*" in kcat, "*K*" in KM)

*P3, L50 Should read "threats"

*P5, L134: should read "MHET"

*P5, L115:

"Although CALB has been scarcely employed for degrading PET, we envision CALB as a promising chassis to design a highly efficient PETase due to its activity promiscuity, the architecture of its active site and its thermal stability."

I don't understand this part, CalB has a buried active site, how could it degrade a polymer?

*conclusion: "dimers", please correct

Reviewer #2 (Remarks to the Author):

Swiderek et al in the current manuscript report an interesting mechanistic study on the effect of pH on the selectivity of lipase by both experimental and computational methods. The changes in the protonation state of a number of residues under pH=5 and pH=9 were proposed to be the key to the differences in the experimentally observed selectivity. The complete reaction pathways of the hydrolysis of BHET were investigated. Moreover, the pH-dependent selectivity of the hydrolysis of BHET, which was obtained from PET by chemical depolymerization, was also explored. From my point of view, this work constitutes an important contribution to the understanding of lipase-catalyzed degradation. However, the following concerns should be properly addressed before publication.

1. On the basis of the computational results, the authors proposed some key residues in affecting the selectivity of the reaction. Mutation analysis of these residues should be made to verify the proposal.

2. The energy profiles of the lipase-catalyzed hydrolysis of MHET should be also calculated since the inactivity under pH=9 might be a result of the high barrier of the chemical steps (in addition to the disfavored binding).
3. The effect on the energetics by including the residues making up the oxyanion hole in the QM region should be tested.

Reviewer #3 (Remarks to the Author):

Swiderek and colleagues describe the influence of buffer pH on the hydrolysis of the terephthalic acid (TPA) di-ester (BHET) to either the monoester (MHET) or free acid (TPA). It is interesting that for this enzyme product selection can be determined in such a simple way.

However, the title and the content of the abstract, which both refer to “the hydrolysis of PET oligomers”, are rather misleading. In the field, the term “PET oligomers” refers to chains of *multiple* terephthalic acid moieties linked by ethylene glycol. There are no experiments in this manuscript (or the supplement) that use such substrates. Instead, the authors describe the pH-dependent enzymatic hydrolysis of a di-ester of terephthalic acid, i.e. BHET. No one in the PET field would consider this to be an “oligomer”. Consequently, I would expect this manuscript to be of far lower interest to the PET recycling field than if the enzyme were genuinely depolymerising PET oligomers.

In my opinion, the introduction is overly long and could be significantly shorter without losing the emphasis of the paper (e.g. the first paragraph is far removed from the research topic).

I am not an expert in computational analyses, so I shall restrict the rest of my comments to the other experimental work.

In the final paragraph (p.5), the authors claim: “The potential of this pH-controlled biotransformation is illustrated by the valorisation of BHET... ..to yield MHBET [sic], an intermediate that is very hard to achieve through chemical methods.” (The abbreviation MHBET does not appear anywhere else in the paper, so I assume this is supposed to read “MHET”.) This is simply not true. The authors’ own data (Fig. 5b) shows the susceptibility of BHET to alkaline hydrolysis (at pH 9) to yield MHET (and no TPA) in the absence of added enzyme; and at pH 10, this reaction would be 10-times faster. Therefore, BHET to MHET conversion is actually very *easy* to achieve with basic chemical methods. Indeed, in the Results and Discussion the authors contradict themselves by stating “...at pH 9, BHET was spontaneously hydrolysed to MHET...”.

The k_{cat}/K_m numbers in Table 1 lack error values. These could have easily been calculated from the error values provided for the individual k_{cat} and K_m values. Some of these k_{cat}/K_m errors are actually quite large, for example I calculate that the k_{cat}/K_m of 0.067 /Ms for BHET at pH 7 has an error of 0.040 /Ms. This will impact on the statistical significance of any comparison between them. I note also a lack of p-values in comparing enzymatic efficiencies.

Similarly, the authors should recognize and highlight that they have extrapolated K_m values well beyond the upper limit of maximum substrate concentration, which for BHET 10 mM was 10 mM (as stated in the Methods). I would have expected this to be stated in the footnotes of Table 1.

There are significant technical shortcomings in the hydrolysis experiments. Given its pKa values (2.1, 7.2 and 12.4), phosphate is a particularly poor choice of buffer for reactions performed at pH 5 and pH 9, even at 100 mM buffer. This is particularly problematic for BHET hydrolysis which releases two H⁺ upon conversion to TPA, so complete hydrolysis of the 10 mM BHET would have released 20 mM H⁺. I note that the authors do not report the pH change at the end of the reactions. If the reactions had been done in a pH-controlled reactor, then this would not have been such an issue.

Also, in the data shown in Figure 5, it is not clear how the BHET and MHET concentrations could be 12-14 mM when the starting BHET concentration was 10 mM. This appears to be a calibration problem from the UHPLC analysis.

In summary, despite the technical shortcomings, I think this is an interesting study from an enzyme substrate-dependence point of view, but regrettably I don't think it is particularly significant for the field of plastics recycling/upcycling.

Prof. Vicent Moliner, Fellow of the RSC
Institute of Advanced Materials (INAM)
Universitat Jaume I, 12071 Castellón, Spain
Telephone +34 964728084
Email: moliner@uji.es
www.biocomp.uji.es

RESPONSE TO THE REVIEWERS

Reviewer's comments in black

Authors' responses are in blue, the same as the key changes in the manuscript.

Reviewer #1 (Remarks to the Author):

The manuscript by Świderek et al. is centered on the coherent topic of biotechnological-assisted plastic recycling. The study is focused on *Candida antarctica* lipase B (CalB) and describes its ability to degrade BHET (obtained from chemical degradation of PET) into either MHET (monoacid) or TPA (diacid). The authors describe a selectivity switch with respect to formation of either MHET or TPA from BHET by CalB at acidic or basic pH, and use computational tools (MD and hybrid QM/MM techniques) to shed mechanistic light on the observed switch. I find the topic of the manuscript interesting and the computational work seems well performed.

R: We appreciate the comments of reviewer 1 and we agree about the high potential interest that the presented topic can provide for the journal's readers. In fact, we have incorporated his/her clear definition of "selectivity switch" in the Conclusions section.

Still, I have concerns:

Major:

*Novelty: The authors make arguments that their study would enable isolation of either TPA or MHET, with MHET being a difficult to access intermediate by chemistry. In fact, a report by Eugenio et al. have already described the use of hydrolases for efficient biosynthesis of MHET from BHET (10.1016/j.jbiotec.2022.08.019). Likewise, specificity switch for either MHET or BHET as substrate is known (<https://doi.org/10.1038/s41467-019-09326-3>). In the light of these previous report, my opinion is that the present contribution would be more suitable for a specialized journal.

R: With all our respect to reviewer #1, we do not think these cited papers already describe or find the results and conclusions of our study. The paper of Eugenio et

al. describes the effect of the substrate concentration on the kinetic of the selective hydrolysis of BHET to MHET, but it does not demonstrate that this enzyme can do a single hydrolysis or two sequential hydrolysis depending on the reaction conditions (i.e pH as in our case). We find an unprecedented enzyme selectivity controlled by the reaction pH. Our findings are indeed very well summarized by this reviewer, as mentioned above: “The authors describe a selectivity switch with respect to the formation of either MHET or TPA from BHET by CALB at acidic or basic pH, and use computational tools (MD and hybrid QM/MM techniques) to shed mechanistic light on the observed switch.”; and by reviewer #3: “Swiderek and colleagues describe the influence of buffer pH on the hydrolysis of the terephthalic acid (TPA) di- ester (BHET) to either the monoester (MHET) or free acid (TPA). It is interesting that for this enzyme product selection can be determined in such a simple way.”

The other paper (Palm et al. Nat. Commun. (2019)10:1717 <https://doi.org/10.1038/s41467-019-09326-3>) cited by the reviewer to support the lack of novelty of our work, reports the crystal structures of active ligand-free MHETase and MHETase bound to a non-hydrolyzable MHET. The activity of PET hydrolases has been described in the past, as commented in our manuscript. It is however very interesting that in this paper of Palm et al. the authors also report kinetic studies of turnover of BHET and MHET with MHETase mutants, showing different activities depending on the specific variants. However, these mutants were never tested with a reaction mixture where both BHET and MHET co-exist. In contrast, our study shows, as mentioned by reviewer #3, a very simple way to control in situ the product selectivity by just changing the pH, without further protein engineering. We appreciate that the reviewer mention that “...(we)... use computational tools (MD and hybrid QM/MM techniques) to shed mechanistic light on the observed switch. I find the topic of the manuscript interesting, and the computational work seems well performed”. Aligned with this comment, we remark on the importance of computational simulations based on high-level QM/MM MD simulations to unveil the pH-driven switch on CALB selectivity. Therefore, the computational study was not only used to describe the mechanism but to predict the effect of the pH on the selectivity of an enzyme that has not been previously used to degrade BHET or MHET.

We have introduced and commented on the paper of Palm et al. in the Introduction section because of its relevance in the field (page 3, ref 12). We thank the reviewer for calling our attention to this study which is definitely related to our study.

*Experiments and kinetics. What is the solubility of MHET, BHET and TPA under the experimental conditions, and how is the solubility of the compounds affected by the different pH-values used?

R: The solubility of substrates is an important consideration raised by reviewer 1. To this concern, the reported water solubility of the more hydrophobic compound, BHET, is $8 \text{ g} \times \text{L}^{-1}$ at $25 \text{ }^\circ\text{C}$ (Patent EP0723951A1). In contrast, our reaction mixtures are performed at $2.54 \text{ g} \times \text{L}^{-1}$ (corresponding to 10 mM). Hence, we are below the solubility limit of BHET using 10 mM. To assure that BHET was fully soluble during the whole biotransformation, we added 10% of DMSO as co-solvent

as it has been reported that organic solvents increase the BHET solubility up to 100-fold (<https://doi.org/10.1016/j.cjche.2021.03.024>). In the case of TPA, the water solubility at 25° C is as low as 0.017 g x L⁻¹ (0.1 mM), however, it increases up to more than 1 M just by adding 1% DMSO (<https://doi.org/10.1021/je049577c>), therefore the solubility of TPA in the reaction media is guaranteed under our conditions (10% DMSO and 10 mM TPA).

Similarly, MHET has proven soluble in water up to 60 mM at pH 7 and 40 °C (<https://doi.org/10.1016/j.jbiotec.2022.08.019>), supporting the fact that neither are we suffering solubility issues of this monoester. Regarding the effect of the pH on their solubility, the reported water solubility of MHET decreases roughly 5 times from pH 7 to pH 5 (<https://doi.org/10.1016/j.jbiotec.2022.08.019>). Although MHET is more water soluble at alkaline pH, our study demonstrates that CALB does not hydrolyze MHET at pH 9, whereas it does at pH 5 where its solubility is lower. These results clearly demonstrate that the enzymatic hydrolysis of MHET at pH 5 is due to the protonation state of titratable residues of the CALB enzyme at acid conditions rather than to substrate solubility changes at the different pHs.

A hint to the fact that solubility might be an issue is that NMR shows formation of TPA, whereas this was not observed by UPLC-MS, and I assume the workup was different for these methods? (did not find enough details in the experimental section to fully evaluate/understand how the experiments were performed)

As suggested by the reviewer we added to the materials and methods sections more details about the sample preparation for UPLC and NMR analysis (Pag. 18, bottom). We also detected TPA in the reaction at pH 9 in the UPLC chromatograms as well as in the NMR spectra when we used crude BHET coming from the depolymerization reactions. This is clearly illustrated by figures 5B and S8 where a TPA concentration lower than 1 mM was detected but to a much lower extent than MHET. NMR is only indicating the presence of TPA, but for quantification purposes we rely on the UPLC data. We do not believe that the detection of TPA in NMR is due to a higher solubility of TPA under the workup conditions of NMR because reaction samples were processed through exactly the same methodology at both pH 5 and 9.

*Figure 5c: how was the “yield” quantified?

R: As requested by reviewer 1, we detailed the yield calculation in the Methods section, BHET hydrolysis section (Pag. 18).

*What is the impact of ionic strength on the observed results?

R: In all reactions, the buffer concentration was 100 mM keeping the pH constant during the whole biotransformation. The initial BHET concentration was 10 mM, hence the maximum expected TPA or MHET concentration is the same (10-times lower than the ionic strength of the buffer employed). This ionic strength does not negatively affect the biotransformation since CALB activity negligibly changes between 10 and 100 mM buffer concentration (i.e sodium phosphate). We need to use a high concentration of buffer to avoid pH changes during the hydrolysis of

BHET to either TPA or MHET, which would directly affect the product profile of the enzymatic hydrolysis of BHET. In the methods, we have stressed the necessity of a high ionic strength buffer to maintain the reaction pH constant (Pag. 18, BHET hydrolysis section).

*Protonation states: The authors state that MHET is negatively charged at pH 5 and that its pKa resides between 3.5-4.8. If the pKa is close to 5, it means that half of the available substrate would be protonated under the acidic conditions. How is this accounted for in the study? Also, how did the authors take into account the fact that the catalytic histidine will change its protonation state in the pH range studied, and how would that impact the selectivity?

R: We agree with the reviewer that the determination of the protonation state of the titratable residues is a crucial step when setting up a molecular model. For this reason, an exhaustive study was carried out not only by means of the semiempirical PropKa method but also by computing the full titratable curves of all titratable residues based on constant-pH hybrid neMD/MC simulations (as explained in the Computational Details section). Regarding the catalytic histidine His224, obviously, its protonation state is changing as a consequence of the pH. However, we have to assume an initial protonation state for the chemical reaction to take place in the active site. This particular pKa is modulated along the reaction by specific direct interactions such as those of Asp187 that tune the pKa of His224, as shown in previous studies. Regarding the substrate, MHET, the reviewer is right that a non-negligible fraction can be protonated at pH 5 but, keeping in mind that the accepted pKa of this molecule (3.5 - 4.8, depending on the substrate configuration within the active site and its local environment as stated by Park et al., ref 36) is below pH 5, we can accept that most of the substrate molecules will be deprotonated. We assume that the dramatic effect of the pH change between 5 and 9 will be on the titratable protein residues, and not so much on the substrate. The computer simulations based on this assumption allow for predicting and explaining the experimental results, which supports our complete proposal.

*Fig S6, crude NMR. What are the unassigned peaks?

R: Unassigned peaks in figure S6 can be attributed to ethyleneglycol formation (signal at 3.6 ppm), whereas DMSO is also visible (signal at 2.5 ppm). The peaks were assigned in the revised version of Fig. S6.

Minor and editorial comments:

*please use italics for kinetic constants ("*k*" in kcat, "*K*" in KM)

R: In agreement with the comment, we have used italics for all kinetic constants.

*P3, L50 Should read "threats"

*P5, L134: should read "MHET"

*conclusion: "dimers", please correct

R: We acknowledge the corrections. We have addressed them in the manuscript.

*P5, L115: "Although CALB has been scarcely employed for degrading PET, we

envision CALB as a promising chassis to design a highly efficient PETase due to its activity promiscuity, the architecture of its active site and its thermal stability.” I don’t understand this part, CalB has a buried active site, how could it degrade a polymer?

R: CALB is the most prominent promiscuous lipase exhibiting activity over a wide variety of monomeric and polymeric substrates. Good examples of this catalytic ability include the degradation of aliphatic polyesters such as PLLA, PBA, PBS, PCL, PHB, PTMC and PTCL (<https://doi.org/10.1007/s10924-017-0945-1>). The substrate promiscuity displayed by this enzyme is mainly related to its catalytic triad and the plasticity of the catalytic pocket which is covered by a small hydrophobic lid remaining in an open conformation in contact with hydrophobic substrates (<https://doi.org/10.1128/9781555816827.ch36>).

We have cited this article (<https://doi.org/10.1007/s10924-017-0945-1>, ref 17) in page 4 and 5 to support the potential of CALB in polyester depolymerization.

Reviewer #2 (Remarks to the Author):

Swiderek et al in the current manuscript report an interesting mechanistic study on the effect of pH on the selectivity of lipase by both experimental and computational methods. The changes in the protonation state of a number of residues under pH=5 and pH=9 were proposed to be the key to the differences in the experimentally observed selectivity. The complete reaction pathways of the hydrolysis of BHET were investigated. Moreover, the pH-dependent selectivity of the hydrolysis of BHET, which was obtained from PET by chemical depolymerization, was also explored. From my point of view, this work constitutes an important contribution to the understanding of lipase-catalyzed degradation.

R: We appreciate the comment of reviewer 2 and agree with the importance of unveiling an unprecedented pH-control on the selective hydrolysis of polyesters.

However, the following concerns should be properly addressed before publication.

1. On the basis of the computational results, the authors proposed some key residues in affecting the selectivity of the reaction. Mutation analysis of these residues should be made to verify the proposal.

R: We agree with the comment of reviewer 2 about the interest that a mutation analysis might have. However, the suggested work will suppose a huge experimental and computational effort that is out of the scope of this article. We prefer to maintain the readers’ focus on the present study where only the wild-type CALB is employed and the effect of the pH is revealed as a possible selectivity switch, as mentioned by the reviewers. We acknowledge the comments of the reviewer and, following his/her suggestion, we agree that future studies focused on confirming the role-specific residues should be carried out in our laboratories.

2. The energy profiles of the lipase-catalyzed hydrolysis of MHET should be also

calculated since the inactivity under pH=9 might be a result of the high barrier of the chemical steps (in addition to the disfavored binding).

R: We agree with the reviewer that pH could also affect the chemical steps of the lipase-catalyzed hydrolysis of MHET, as we have demonstrated that it affects the hydrolysis of BHET, especially in the acylation step (see Fig.S19, S20 and the resulting free energy profiles in Fig. 4). However, as the reviewer points out, we already observed a dramatic unfavorable effect on the binding of MHET that suggest that the substrate will hardly bind the active site at pH 9. If accepting that MHET could be properly posed in the active site as the BHET does, we can accept that the chemical reaction steps will not dramatically differ from each other. This is confirmed by the so close activation free energies derived from the experimentally determined rate constants of the hydrolysis of BHET and MHET at pH 5, as listed in Table 1 (17.8 vs 17.3 kcal·mol⁻¹). Finally, it is equally important to point out that the high-level simulations carried out in the present study, with the fully flexible solvated system represent enormous computational time.

Following the reviewer's suggestion, we have added some comments on pages 10-11 to justify our strategy and to clarify this point to the readers.

3. The effect on the energetics by including the residues making up the oxyanion hole in the QM region should be tested.

R: The size of the QM region can have a dramatic effect on QM/MM results, as proposed by the reviewer. However, this is not the first study we carried out on CALB reactivity (and other enzymatic systems where the substrate is partially anchored by the presence of an oxyanion hole in the active site), as demonstrated by the related papers cited in the manuscript. Our experience demonstrates that enlarging the QM region by including the residues that participate in the oxyanion hole does not provide a significant improvement in the results. In contrast, enlarging the QM region not only dramatically increases the cost of the simulations but also forced the introduction of more link atoms which is always a possible source of error. Finally, keeping in mind that the interaction between the oxyanion hole and the O ϵ 1 atom of the substrate is basically electrostatic in nature, our assumption that not a significant charge transfer is taking place between the protein and the substrate (by describing the oxyanion hole by classical force fields and the substrate by a quantum Hamiltonian) looks an accepted assumption.

Reviewer #3 (Remarks to the Author):

Swiderek and colleagues describe the influence of buffer pH on the hydrolysis of the terephthalic acid (TPA) di-ester (BHET) to either the monoester (MHET) or free acid (TPA). It is interesting that for this enzyme product selection can be determined in such a simple way. However, the title and the content of the abstract, which both refer to “the hydrolysis of PET oligomers”, are rather misleading. In the field, the term “PET oligomers” refers to chains of *multiple* terephthalic acid moieties linked by ethylene glycol. There are no experiments in this manuscript (or the supplement) that use such substrates. Instead, the authors describe the pH-dependent enzymatic hydrolysis of a di-ester of terephthalic acid, i.e. BHET. No one in the PET field would consider this to be an “oligomer”. Consequently, I would expect this manuscript to be of far lower interest to the PET recycling field than if the enzyme were genuinely depolymerising PET oligomers.

R: First, we apologize if our title misled reviewer 3. We agree with her/him that the BHET is a diester of terephthalic acid rather than a PET oligomer. For this reason, we have done an effort to demonstrate the selectivity switch of CALB driven by the pH using a PET trimer obtained from the organocatalytic depolymerization of PET plastics. The results, now added to the manuscript and Supplementary Fig S9, confirm that the selectivity switch also works for PET oligomers beyond the model BHET substrate. These results can be used for the valorisation of BHET and PET oligomers resulting from the organocatalytic depolymerization of PET to produce either MHET or TPA depending on the reaction pH.

In my opinion, the introduction is overly long and could be significantly shorter without losing the emphasis of the paper (e.g. the first paragraph is far removed from the research topic).

R: We agree with the reviewer and, consequently, the introduction has been shortened.

I am not an expert in computational analyses, so I shall restrict the rest of my comments to the other experimental work.

In the final paragraph (p.5), the authors claim: “The potential of this pH-controlled biotransformation is illustrated by the valorisation of BHET... ..to yield MHBET [sic], an intermediate that is very hard to achieve through chemical methods.” (The abbreviation MHBET does not appear anywhere else in the paper, so I assume this is supposed to read “MHET”.) This is simply not true. The authors’ own data (Fig. 5b) shows the susceptibility of BHET to alkaline hydrolysis (at pH 9) to yield MHET (and no TPA) in the absence of added enzyme; and at pH 10, this reaction would be 10-times faster. Therefore, BHET to MHET conversion is actually very *easy* to achieve with basic chemical methods. Indeed, in the Results and Discussion the authors contradict themselves by stating “...at pH 9, BHET was spontaneously hydrolysed to MHET...”.

R: We appreciate the correction of the abbreviation of MHET, which was wrongly written as MHBET. However, with all the respect to reviewer 3, we disagree with his/her major criticism. This work presents valuable results which can be used for the valorisation of BHET, but also of PET oligomers resulting from the organocatalytic depolymerization of PET to produce MHET (as demonstrated by our new results added into the manuscript). Although MHET can be achieved through chemical methods under alkaline pH using strong bases, CALB means a more sustainable alternative for the selective transformation of PET oligomers into MHET. As stated in the abstract and the conclusions, we demonstrate for the first time the selective biotransformation of PET oligomers and diesters based on the pH of the reaction media. This discovery allowed us to perform the selective hydrolysis of BHET, but also of PET trimers, to either its corresponding diacid or monoester using both soluble and immobilized CALB just by tuning the pH of the reaction. These results can be used for the valorisation of BHET resulting from the organocatalytic depolymerization of PET to produce MHET.

The k_{cat}/K_M numbers in Table 1 lack error values. These could have easily been calculated from the error values provided for the individual k_{cat} and K_M values. Some of these k_{cat}/K_M errors are actually quite large, for example I calculate that the k_{cat}/K_M of 0.067 /Ms for BHET at pH 7 has an error of 0.040 /Ms. This will impact on the statistical significance of any comparison between them. I note also a lack of p-values in comparing enzymatic efficiencies.

R: We appreciate reviewer 3's suggestion. Accordingly, we have included the error values of k_{cat}/K_M as well as the corresponding ANOVA analysis with a $P < 0.05$ to endorse the statistical significance of the data presented in Table 1 (Pag. 7).

Similarly, the authors should recognize and highlight that they have extrapolated K_M values well beyond the upper limit of maximum substrate concentration, which for BHET 10 mM was 10 mM (as stated in the Methods). I would have expected this to be stated in the footnotes of Table 1.

R: Following the recommendation of reviewer 3, we have included this information in the footnotes of Table 1 (Pag. 7).

There are significant technical shortcomings in the hydrolysis experiments. Given its pKa values (2.1, 7.2 and 12.4), phosphate is a particularly poor choice of buffer for reactions performed at pH 5 and pH 9, even at 100 mM buffer. This is particularly problematic for BHET hydrolysis which releases two H^+ upon conversion to TPA, so complete hydrolysis of the 10 mM BHET would have released 20 mM H^+ . I note that the authors do not report the pH change at the end of the reactions. If the reactions had been done in a pH-controlled reactor, then this would not have been such an issue.

R: We want to apologize for this mistake in the Figure 5 caption. We only used phosphate buffer for reactions performed at pH 7. For pH 5 and pH 9, we used sodium acetate and sodium bicarbonate buffers, respectively to properly maintain the pH. This was indicated in the BHET hydrolysis section (Pag. 16) of the original

submission and corrected in the Figure 5 caption of the revised submission (Pag. 14).

Also, in the data shown in Figure 5, it is not clear how the BHET and MHET concentrations could be 12-14 mM when the starting BHET concentration was 10 mM. This appears to be a calibration problem from the UHPLC analysis.

R: We want to apologize for this mistake in the Y axis legends. It was an error transcribing the dataset from the spreadsheet of the raw data to the Origin software which we create the plots with. We have inserted the correct Y-axis legend in Figs. 5a, 5b and S3.

In summary, despite the technical shortcomings, I think this is an interesting study from an enzyme substrate-dependence point of view, but regrettably I don't think it is particularly significant for the field of plastics recycling/upcycling.

R: We agree that the original employed substrate in our study was a diester of terephthalic acid, i.e. BHET, which is not what is considered as oligomer in the field of polymers. However, to support the potential of this work in the field of plastic recycling/upcycling, we have intended the selective hydrolysis of PET trimers under different pH conditions. The PET trimers generated from the organocatalytic depolymerization of PET were incubated with CALB at pH 5 and 9, confirming the selectivity switch observed in these enzymes for BHET but now for the hydrolysis of PET oligomers. This new data supports the potential of CALB in the depolymerization of PET plastics. This is one of the few works where a two-pot chemo and biocatalytic cascade reaction work to transform PET into small monomers/oligomers that can be further valorized (either TPA or MHET). Remarkably, this has been possible through an interdisciplinary collaboration that has allows an understanding of the mechanisms of CALB to control its selectivity to transform the products of the organocatalytic depolymerization in valuable monomers under mild conditions. We are aware that the last developments in engineered PETase are boosting the field of biocatalytic plastic degradation however these engineered enzymes are still far from the processability numbers the industry demands. In this context, we believe that plastic degradation will become a reality only when chemo(enzymatic) approaches, like the one we herein present, are deployed. We stressed this point in the last sentence of the conclusion section of the revised manuscript.

REVIEWER COMMENTS

Reviewer #1 (Remarks to the Author):

The authors did a nice and convincing job in addressing most of my comments. I still have concerns about the following points that the authors should address:

*Ionic strength (same comment as before):

A sodium acetate buffer will have a different ionic strength compared to a phosphate buffer of the same molarity. To account for this, experiments should be done at the same ionic strength. I appreciate that the authors state that the difference in activity between 10 mM and 100 mM phosphate buffer is small (although I had some difficulties in finding this data in the MS). Is the selectivity also the same at different ionic strengths?

*Protonation states: Here I have the same comment as before (The authors state that MHET is negatively charged at pH 5 and that its pKa resides between 3.5-4.8. If the pKa is close to 5, it means that half of the available substrate would be protonated under the acidic conditions).

The authors should (computationally) show that a protonated MHET would not explain the selectivity switch. I think this is a key point - and in my opinion a remaining weakness of the study

Reviewer #2 (Remarks to the Author):

The authors have replied to my concerns with explanations rather than extra experimental and computational works. However, I do think all of those points are very important in improving the quality of the present manuscript. And more importantly, some of them actually cannot be ignored since they might change the conclusions of the current work.

1. The suggested verification of prediction indeed requires extra work. However, in my opinion, the results would provide strong support to the prediction and are necessary to perform. Furthermore, as the present work is already a combined experimental and computational work, the efforts for the required mutation analysis is not that huge.

2. I agree with the authors that the mechanism might be the same for the hydrolysis of MHET, however, I still recommend the authors to verify this by further calculations, rather than leave it as an open question.

3. Although the authors mentioned “Our experience demonstrates that enlarging the QM region by including the residues that participate in the oxyanion hole does not provide a significant improvement in the results.”, however, as a computational chemist, I have experienced the cases that the effect of oxyanion hole is significant. Considering that the QM region in the QMMM simulations is not big, I still strongly recommend the authors to test the suggestion.

Reviewer #3 (Remarks to the Author):

In this revised manuscript, Swiderek and colleagues describe the influence of buffer pH on the products generated from the hydrolysis of the terephthalic acid (TPA) di-ester, BHET. As I commented previously, it is interesting that, for this enzyme, product selection can be determined in such a simple way. I am pleased to see that the authors have taken the majority of my comments on board, included additional experimental data, and revised the manuscript accordingly. The addition of the hydrolysis experiments on PET oligomers are particularly welcome.

Therefore, I am happy to recommend the revised manuscript for publication in Nature Communications.

RESPONSE TO THE REVIEWER

Reviewer's comments in black

Authors' responses are in blue, the same as the key changes in the manuscript.

Reviewer #1 (Remarks to the Author):

The authors did a nice and convincing job in addressing most of my comments. still have concerns about the following points that the authors should address:

R: We appreciate the positive comment of the reviewer about the additional job we did to address his/her concerns. In fact, we acknowledge his/her previous suggestions that have contributed to increasing the quality of the study. Regarding the remaining issues, they have been addressed, as follows.

*Ionic strength (same comment as before):

A sodium acetate buffer will have a different ionic strength compared to a phosphate buffer of the same molarity. To account for this, experiments should be done at the same ionic strength. I appreciate that the authors state that the difference in activity between 10 mM and 100 mM phosphate buffer is small (although I had some difficulties in finding this data in the MS). Is the selectivity also the same at different ionic strengths?

R: The data about the effect of the ionic strength on the CALB activity were not included in the revised version of the manuscript. We considered their relevance very low in the context of the data we presented since low ionic strength was never used for any of the shown experiments. Nonetheless, we have now included this data in the supporting information (Fig. S3a, including the description of the new panel in the caption) and one sentence was added to the revised manuscript (page 6).

Regarding the effect of the ionic strength on the selectivity, it is very hard to experimentally prove it. If we decrease the buffer concentration (lower ionic strength) for the reactions at pH 7 and pH 9, the pH drops and therefore the selectivity of the enzyme changes because of the pH rather than by the ionic strength. In the case of pH 5, at low ionic strength, the pH goes to very acidic values inactivating the enzyme. Therefore, we suggest that both ionic strength and pH are two parameters difficult to be separately analyzed during the course of the reaction.

* *Protonation states: Here I have the same comment as before (The authors state that MHET is negatively charged at pH 5 and that its pKa resides between 3.5-4.8. If the pKa is close to 5, it means that half of the available substrate would be protonated under the acidic conditions).

The authors should (computationally) show that a protonated MHET would not explain the selectivity switch. I think this is a key point – and in my opinion a remaining weakness of the study

R: Following the reviewer's suggestion, and considering that a non-negligible fraction of the MHET population can be protonated at pH 5 as stated by the reviewer, we have prepared a new model with neutral MHET and we have

repeated the *in silico* test based on unconstrained Molecular Dynamics simulations to explore whether a protonated MHET remains in the active site or not. The results are reported and discussed in the new version of the manuscript (see pages 9-10 and new Fig. 3, left panel). These results confirm that the protonated MHET is also stable in the active site of CALB at pH 5. This new result provides additional value to the computer simulations and gives robustness to the conclusions. We acknowledge his/her suggestion.

Reviewer #2 (Remarks to the Author):

The authors have replied to my concerns with explanations rather than extra experimental and computational works. However, I do think all of those points are very important in improving the quality of the present manuscript. And more importantly, some of them actually cannot be ignored since they might change the conclusions of the current work.

R: We appreciate the time of reviewer 2 in reviewing our manuscript. However, we respectfully disagree with him/her in the sense that we do not (and we did not) consider that the suggested additional calculations could change the message and conclusions of our study. We agree with him/her that some of his/her suggestions can be interesting to improve the robustness of the present study but not necessary to change the main message. Anyway, as described below point by point, we have carried out additional calculations following his/her suggestions and, as commented below, addressing the first two comments has allowed improving the quality of the manuscript, which we deeply acknowledge.

1. The suggested verification of prediction indeed requires extra work. However, in my opinion, the results would provide strong support to the prediction and are necessary to perform. Furthermore, as the present work is already a combined experimental and computational work, the efforts for the required mutation analysis is not that huge.

R: As we already mentioned in our previous letter, we agree with the comment of the reviewer about the interest that a mutation analysis might have. However, we disagree with the reviewer in the sense that the suggested work “is not that huge”. As he/she knows, performing *in silico* mutations on a protein implies revising the model and carrying out long MD simulations to equilibrate the new system. On the other side, an experimental validation, as commented in the previous answer to the reviewer, would represent a huge additional experimental effort that is out of the scope of this study, where we prefer to maintain the readers’ focus on the study on the wild-type CALB, and the effect of the pH that is revealed as a possible selectivity switch, as highlighted by all the reviewers.

However, and taking into account that the additional simulations suggested by the reviewer can provide additional robustness to our hypotheses to explain the effect of the pH, following the reviewer’s suggestion we have performed the *in silico* mutation of D134A to confirm the role of D134. As observed in Fig.3, a repulsion is established between the negatively charged D134 and deprotonated MHET at pH 9. Mutation of D134 to Ala restores the favorable interactions between MHET⁽⁻⁾ and the active site, and therefore MHET remains in the pocket (see red line in the new Fig. S19). This *in silico* mutation proves our hypothesis on the role of D134. These new results (reported in Fig. S19) are discussed in the new version of the manuscript (see page 10). We thank the reviewer for his/her comment.

2. I agree with the authors that the mechanism might be the same for the hydrolysis of MHET, however, I still recommend the authors to verify this by further calculations, rather than leave it as an open question.

R: As we commented in our previous letter and the manuscript, the so close activation free energies derived from the experimentally determined rate constants of the hydrolysis of BHET and MHET at pH 5, as listed in Table 1 (17.8 vs 17.3 kcal·mol⁻¹), and the high similarities of the two reactive centers of the two compounds, supports the assumption of similar chemical steps for MHET and BHET, if both substrates remain stable in the active site at the Michaelis complex state. We do not think we were leaving it as an open question. The reviewer knows that the high-level simulations carried out in the present study (2D free energy surfaces, (FES), computed with QM/MM MD for 4 chemical steps), with the fully flexible solvated system represent enormous computational time if the full reaction mechanism should be explored with the new substrate, MHET.

Anyway, in order to confirm the assumption of an equivalent reaction mechanism for MHET and BHET hydrolysis catalyzed by CALB, and following the reviewer's suggestion, we have repeated the exploration of the full reaction with MHET at pH 5, conditions under which the reaction is feasible as suggested by the experimental data and the MD simulations. The new results, achieved by temporarily devoting all our computer resources to this task, confirm that the molecular mechanism does not depend on the substrate, BHET vs MHET. The new FESs of MHET hydrolysis deposited in the Supporting Information (new Fig. S22) are virtually the same as those for the hydrolysis of BHET (see Fig. S20 and S21). In addition, transition state structures for all steps of the hydrolysis of MHET were optimized at DFT/MM level, from structures selected from the quadratic regions of the FES (see new Table S8). Frequencies analyses (confirming only one imaginary frequency in the TSs) and IRCs traced down to the associated minima, all data confirming a molecular mechanism same as the one for the hydrolysis of BHET. Energetic data derived from the FESs are listed in Table S3, rendering a value of the free energy of activation of 18.3 kcal·mol⁻¹ for the rate-limiting step, quite close to those for the hydrolysis of BHET (18.0 and 17.1 kcal·mol⁻¹ for the hydrolysis of BHET at pH 5 and 9, respectively), in agreement with the experimental data. Some comments have been added to the text (see pages 11 and 12). We thank the reviewer for his/her suggestion which allow providing additional validation to our predictions.

3. Although the authors mentioned "Our experience demonstrates that enlarging the QM region by including the residues that participate in the oxyanion hole does not provide a significant improvement in the results.", however, as a computational chemist, I have experienced the cases that the effect of oxyanion hole is significant. Considering that the QM region in the QMMM simulations is not big, I still strongly recommend the authors to test the suggestion.

R: As we commented in our previous letter, the size of the QM region can have a dramatic effect on QM/MM results, as discussed by the reviewer. However, this is not the first QM/MM study we carried out on CALB reactivity (and other enzymatic systems where the substrate is partially anchored by the presence of an oxyanion hole in the active site). We also mentioned in our previous letter that enlarging the QM region not only dramatically increases the cost of the simulations but also forces the introduction of more link atoms which is always a source of error in

QM/MM based calculations. Finally, keeping in mind that the interaction between the oxyanion hole and the O3 atom of the substrate is basically electrostatic in nature, a negligible change of charge transfer between the protein and the substrate along the reaction is a reasonable assumption. It is important to point out that the QM/MM scheme we are using is an *additive scheme*, and not a *subtractive scheme* such as the one used with, for instance, the ONIOM program originally developed by Morokuma and co-workers. Thus, the interaction between the QM region and the MM region is computed at the QM level, which explains why many authors employ this partition. For instance, as mentioned, we have tested and applied this QM-MM partition in all our studies on the reactivity of CALB (ie. ACS Catal. 2020, 10, 1938–1946, ACS Editors' Choice; ACS Catal. 2021, 11, 8635–8644; J. Chem. Inf. Model. 2021, 61, 3604–3614; Chem. Sci. 2022, 13, 4779–4787) and other enzymatic systems including a related QM/MM study on the depolymerization of PET by two hydrolases (J. Chem. Inf. Model. 2021, 61, 3041–3051). Moreover, studies published by recognized researchers in this field such as Prof. McCammon focused on the role of the catalytic triad and the oxyanion hole in acetylcholinesterase catalysis by *ab initio* QM/MM methods, treating the residues forming the oxyanion hole classically (J. Am. Chem. Soc. 2002, 124, 10572–10577), or Prof. Mulholland from Bristol University (i.e. ACS Infect. Dis. 2022, 8, 1521–1532) who also uses additive schemes of QM/MM methods to explore chemical reactivity in enzymes with an oxyanion hole in the active site, described at MM level.

Moreover, a recent study on the mechanism and biomass association of glucuronoyl esterase has been published in Nature Communications ((2022)13:1449 <https://doi.org/10.1038/s41467-022-28938-w>) where different sizes of the QM region were studied with QM/MM methods similar to the one employed by us. In none of the tested QM models authors introduce the oxyanion hole and, in fact, when the authors attempted to introduce the oxyanion hole (R268 side chain) in the QM region, they obtained an odd result and they concluded “to continue their investigations with the previous model assembly as the best realistic choice”.

As mentioned above, if there is no charge transfer (and evidently not the possibility of a covalent bond formation) between the QM region and the oxyanion hole, and keeping in mind that the interaction is maintained along the chemical reaction, we do not think that increasing the QM size can render new and improved results. In fact, the good agreement between our computational results and our experimental data (in this but in all the previous studies, some of them cited above) does not suggest that the additional huge amount of calculations could provide improved results. On contrary, increasing the QM region, apart from the necessity of adding new quantum link atoms with the corresponding risk of an additional source of error, can unbalance the full QM region, as occurred in the Nature Communications paper commented above.

Anyway, following the reviewer's suggestion, we have repeated the tests that we already did for the setting up of the QM/MM model of CALB when we first studied this system (ACS Catal. 2020, 10, 1938-1946). We have evaluated whether possible different charge transfers can take place between the oxyanion hole treated at the QM level and the substrate at the different states of the reaction, which would be an indication that the size of the QM region is not large enough. As shown in the table below, there is no significant change in the charge of the

oxyanion hole along the reaction path, thus suggesting no charge transfer that could justify increasing the size of the QM sub-set of atoms. This test supports the originally selected model and confirms that the interactions between the wave function of the QM subset of atoms and the charges of the residues treated as partial point charges are realistically modelled.

ESP atomic charges (in a.u.) computed at M06-2X/MM level with the enlarged QM region (including residues Thr40 and Gly106 in the QM region) on previously optimized geometries along the reaction profile using the CHelpG method.

	RC	TS1	I1	TS2	I2	TS3	I3	TS4	PC
oxyanion hole	-0.162	-0.130	-0.218	-0.203	-0.287	-0.228	-0.323	-0.264	-0.273
rest of the QM system	-0.838	-0.870	-0.782	-0.797	-0.713	-0.772	-0.677	-0.736	-0.727

With all our respect, this is a concern that the reviewer should imagine we are aware of, considering our experience of almost 3 decades developing and applying QM/MM methods to the study of enzyme reactivity.

Reviewer #3 (Remarks to the Author):

In this revised manuscript, Swiderek and colleagues describe the influence of buffer pH on the products generated from the hydrolysis of the terephthalic acid (TPA) di-ester, BHET. As I commented previously, it is interesting that, for this enzyme, product selection can be determined in such a simple way. I am pleased to see that the authors have taken the majority of my comments on board, included additional experimental data, and revised the manuscript accordingly. The addition of the hydrolysis experiments on PET oligomers are particularly welcome. Therefore, I am happy to recommend the revised manuscript for publication in Nature Communications.

R: We appreciate the comments of the reviewer and his/her original suggestions that have allowed improving the quality of our manuscript

REVIEWERS' COMMENTS

Reviewer #1 (Remarks to the Author):

My comments have been addressed.

Reviewer #2 (Remarks to the Author):

Thanks for the authors' effort in the revision. I recommend the publication of the manuscript in the current format.